# The Photocatalytic Degradation of Enrofloxacin Using an Ecofriendly Natural Iron Mineral: The Relationship Between the Degradation Routes, Generated Byproducts, and Antimicrobial Activity of Treated Solutions

**DOI:** 10.3390/molecules29245982

**Published:** 2024-12-18

**Authors:** Sindy D. Jojoa-Sierra, Efraím A. Serna-Galvis, Inés García-Rubio, Maria P. Ormad, Ricardo A. Torres-Palma, Rosa Mosteo

**Affiliations:** 1Grupo de Investigación Agua y Salud Ambiental, Instituto Universitario de Investigación en Ciencias Ambientales de Aragón (IUCA), Universidad de Zaragoza, 50018 Zaragoza, Spain; sindyjojoa@posta.unizar.es (S.D.J.-S.);; 2Grupo de Investigación en Remediación Ambiental y Biocatálisis (GIRAB), Instituto de Química, Facultad de Ciencias Exactas y Naturales, Universidad de Antioquia UdeA, Medellin 050010, Colombia; efrain.serna@udea.edu.co; 3Grupo Catalizadores y Adsorbentes (CATALAD), Instituto de Química, Facultad de Ciencias Exactas y Naturales, Universidad de Antioquia UdeA, Medellin 050010, Colombia; 4Instituto de Nanociencia y Materiales de Aragón (INMA), CSIC-Universidad de Zaragoza, 50009 Zaragoza, Spain; 5Departamento de Física de la Materia Condensada, Facultad de Ciencias, Universidad de Zaragoza, 50009 Zaragoza, Spain

**Keywords:** antibiotic degradation, disinfection, iron oxides, solar photocatalysis, routes and transformations, real-world water treatment

## Abstract

The use of ecofriendly natural minerals in photocatalytic processes to deal with the antimicrobial activity (AA) associated with antibiotics in aqueous systems is still incipient. Therefore, in this work, the capacity of a natural iron material (NIM) in photo-treatments, generating reactive species, to remove the antibiotic enrofloxacin and decrease its associated AA from water is presented. Initially, the fundamental composition, oxidation states, bandgap, point of zero charge, and morphological characteristics of the NIM were determined, denoting the NIM’s feasibility for photocatalytic processes. Consequently, the effectiveness of different advanced processes such as using solar light with the NIM (Light–NIM) and solar light with the NIM and H_2_O_2_ (Light–NIM–H_2_O_2_) to reduce AA was evaluated. The NIM acts as a semiconductor under solar light, effectively degrading enrofloxacin (ENR) and reducing its AA, although complete elimination was not achieved. The addition of hydrogen peroxide (NIM–Light–H_2_O_2_) enhanced the generation of reactive oxygen species (ROS), thereby increasing the elimination of ENR and AA. The role of ROS, specifically O_2_^•−^ and HO^●^, in the degradation of enrofloxacin was distinguished using scavenger species and electron paramagnetic resonance (EPR) analysis. Additionally, the five primary degradation products generated during the advanced processes were elucidated. Furthermore, the relationship between the structure of these products and the persistence or elimination of AA, which was differentiated against *E. coli* but not against *S. aureus*, was discussed. The effects of the matrix during the process and the extent of the treatments, including their capacity to promote disinfection, were also studied. The reusability of the natural iron material was examined, and it was found that the NIM–Light–H_2_O_2_ system showed an effective reduction of 5 logarithmic units in microbiological contamination in an EWWTP and can be reused for up to three cycles while maintaining 100% efficiency in reducing AA.

## 1. Introduction

The safe quality of freshwater is one of the most pressing challenges for both human societies and the natural world, which remains due to the widespread scarcity and pollution of water [1,2]. According to the United Nations, roughly 2.2 billion people lack access to adequately managed drinking water services, attributed to a complex interplay of factors, including climate change, population growth, urbanization, and escalating industrial and agricultural demands [3,4]. Recognizing the necessity to address these issues, recent policies at the European level aimed at reducing harmful discharges into the environment and promoting water reuse as a sustainable solution to manage water resources more efficiently [5,6,7]. In this way, efforts have recently focused on the development of alternatives to conventional treatment for both remediation and water reuse [8,9,10]. As is widely known, in wastewater treatment plants (WWTPs), conventional methods primarily focus on reducing organic pollutants and nutrients like nitrogen and phosphorus through biological processes. While effective in lowering oxygen demand and nutrient levels, these methods often struggle to address emerging contaminants and ensure comprehensive microbiological control in discharged water [11,12,13,14]. Indeed, one of the most concerning issues is the pollution caused by the daily influx of substances with antibiotic activity and microorganisms (with and without antibiotic-resistant genes) into natural aquatic environments [15,16,17].

Antibiotics enter the environment as a result of their widespread use. The annual antibiotic consumption is estimated at approximately 100,000–200,000 tons used worldwide [18,19]. In the health sector, between 2000 and 2015, antibiotic consumption, expressed in defined daily doses (DDDs), increased by 65% (21.1–34.8 billion DDDs), and the rate of antibiotic consumption increased by 39% (11.3–15.7 DDDs per 1000 inhabitants per day), and this trend continues to rise [18]. Recent data show that, in 2022, antibacterial consumption in the European Union was 19.4 DDDs per 1000 inhabitants per day, with country-specific rates ranging from 9.1 to 33.5 DDDs [20]. Thus, the disposal of unused medicines, the excretion of unmetabolized drugs and their metabolites, and pharmaceutical production are important sources of antibiotic contamination from hospitals, industries, and households into wastewater [21,22,23]. Additionally, antibiotic consumption is especially high in countries with large-scale livestock farming, as antibiotics are routinely administered to livestock and aquaculture to prevent and treat infections and also to accelerate growth and increase productivity [24,25]. These practices have become a significant contributor to antibiotic pollution and discharge in water resources and wastewater [26]. According to the World Organization for Animal Health (WOAH), the consumption trends in the last few years are minor compared to projections in the countries monitored by The Global Action Plan on Antimicrobial Resistance, but the demand for animal protein is still expected to rise, resulting in a sustained increase in the use of antibiotics and their discharge in wastewater [27,28].

The prevalence of antibiotics and pathogenic microorganisms in natural ecosystems is particularly concerning in regions where water treatment infrastructure is underdeveloped and insufficient for the populations served. This often results in the direct discharge of wastewater into surrounding areas. Consequently, the detection of antibiotics in drinking water [29,30], surface water [31,32,33], groundwater [34,35], and wastewater is noteworthy. For example, the concentrations of fluoroquinolones and sulfonamide antibiotics such as ciprofloxacin and sulfamethoxazole can reach levels of mgL^−1^ in wastewater and surface water in some African countries, India, and China [33,36,37]. In contrast, European and North American countries typically report concentrations in the range of ngL^−1^ [38,39,40]. Unfortunately, the ubiquity of antibiotics and infective bacteria in the environment can have adverse effects on microbial biodiversity and significant implications for public health, including the proliferation of illness and antibiotic resistance, posing a substantial risk to public health [15,39,41]. The WHO has identified antimicrobial resistance (AMR) as one of the top ten global public health threats facing humanity [42,43,44]. Despite these efforts, drug-resistant infections were estimated to contribute to a devastating 4.95 million deaths globally in 2019. This far exceeds the annual global deaths attributed to tuberculosis (1.5 million), malaria (643,000), and HIV/AIDS (864,000) [44]. Without intervention, it is estimated that global deaths attributable to AMR could reach 10 million annually by 2050 [45]. Thus, strategies for efficiently degrading antibiotics and diminishing their discharge into the environment are required.

In this context, an interesting approach to water treatment involves the use of abundant, cheap, and sustainable resources. Iron oxides and hydroxides, synthetic and natural, have been studied in heterogeneous processes due to their wide diversity, low toxicity, and environmentally friendly characteristics [46,47,48,49]. Furthermore, their use beats several challenges of homogeneous systems, such as catalyst recovery difficulties, the formation of Fe-containing sludges, and secondary pollution caused by the acidic pH conditions required. Potential mechanisms involved in the removal of pharmaceutical drugs using iron-based materials include electrostatic interaction, hydrophobic interaction, hydrogen bond formation, π-interactions, and ion exchange. Additionally, their versatility to act as reducing and oxidizing agents makes them effective for degrading action in catalytic systems, like natural oxidation processes in the environment [50,51,52,53,54]. Particularly, magnetite (Fe_3_O_4_), hematite (α-Fe_2_O_3_), and goethite (α-FeO(OH)) are very effective heterogeneous catalytic systems for eliminating antibiotics such as tetracycline, fluoroquinolones, macrolides, and beta-lactam families [55,56,57,58,59,60,61]. Therefore, research in both controlled and real conditions is crucial for evaluating the feasibility of these methods. However, there is a lack of research that tracks degradation products and their potential impacts. In the context of antibiotic removal, determining their antimicrobial activity (AA) is a critical aspect to consider during water treatment. Beyond the transformation of the initial molecule, analyzing AA allows for the determination of the capacity of processes to eliminate or reduce the risk associated with drug resistance generation. Previous reports have indicated that even when pollutants are thoroughly degraded, the resultant solutions may retain AA [62].

In this work, we considered the treatment of ENR as a model antibiotic. This compound belongs to the therapeutic class of fluoroquinolones, which are among the most consumed antibiotics worldwide that have broad pharmacological spectrum activity used to treat human and veterinary infections, and they are not eliminated by the conventional processes used in municipal wastewater treatment plants. ENR is a highly sold/consumed fluoroquinolone, and it has been detected in the effluents of municipal plants and natural water. Additionally, ENR can induce negative impacts on the environment such as ecotoxicity on plants and the growth inhibition of eukaryotic organisms because of its antimicrobial nature [63]. The remediation of water contaminated with ENR using iron materials has been previously studied. Research has examined the effects of pH and the presence of various anions, cations, and organic materials during the degradation of enrofloxacin using iron oxides [64,65,66,67,68]. Likewise, the degradation of ENR using iron oxides in Fenton processes was recently reported by Sciscenko, I. et al. (2021), who highlighted that the formation of the fluoroquinolones and iron III complex (FQ-Fe(III)) complex decreases the photolytic rate constant of ENR [69], which probably increases the persistence of ENR in the environment. Nevertheless, this coexistence could represent an advantage when achieving Fenton-type treatment at circumneutral pH for the degradation of more recalcitrant pollutants Our previous work explored the optimal parameters for using wüstite, a natural iron material, to eliminate the antibiotic activity of ENR [63], and Wenhui Qiu et al. evaluated the efficacy of iron photo-assisted systems in degrading (UV/Fe^2+^ and UV/H_2_O_2_/Fe^2+^) ENR in water and assessed the toxicity of the resulting solutions on zebrafish embryos at a pH of 3.0 [70]. It is important to mention that the use of ecofriendly natural minerals in photocatalytic processes to deal with the antimicrobial activity (AA) associated with antibiotics in aqueous systems is still scarce. Furthermore, the study of the relationship between the structure of the transformation products coming from antibiotics and the persistence of AA against both Gram-negative and Gram-positive bacteria is not frequently presented in precedent works. Then, the novelty of our research lies in the use of an ecofriendly NIM for the elimination of ENR and decrease in AA, in addition to the determination of the link of the primary transformation products with the antimicrobial activity against two representative microorganisms. Thus, the main goal of the present study was to evaluate the potential of an abundant natural iron material to be utilized in diverse photo-treatments to decrease the presence of antibiotic compounds (such as ENR) in water and the associated antimicrobial activity against Gram-negative (*E. coli*) and Gram-positive (*S. aureus*) bacteria. Herein, an exhaustive study about the determination of reactive species involved in ENR degradation and control subsystems was also carried out. Furthermore, additional tests about inactivation microorganisms, material reuse, and applicability in real water were conducted in this research.

## 2. Results and Discussion

### 2.1. Characterization of Natural Iron Material (NIM)

The elemental analysis of the surface of the NIM was conducted by obtaining X-ray Energy-Dispersive Spectra (EDS). Figure 1a shows that major peaks corresponded to Fe and O, indicating the presence of iron oxides in the NIM. The inset table in Figure 1a lists the minor elements detected, such as C, Ca, Si, and Al, which may have originated from silicates and aluminosilicates in the NIM sample. Additionally, to identify the predominant iron species on both the surface and the bulk of the material (which was conducted by etching), an X-ray photoelectron spectroscopy (XPS) analysis was performed, and the spectrum is shown in Figure 1b. As seen, the XPS spectra of Fe show two intense peaks at 710.7 eV and 724.3 eV, corresponding to the spin–orbit splitting peaks Fe 2p_1/2_ and Fe 2p_3/2_ and shakeup satellite at around 717.2 eV. They are consistent with the presence of Fe^3+^ in the sample. An asymmetric 2p_3/2_ peak and the satellite feature at a higher binding energy could be related to the presence of Fe^3+^ cations at octahedral sites [71,72]. After etching the NIM, there were no substantial differences observed in the XPS profile (Figure 1b), thus indicating a homogeneous composition of the material. The XPS analyses allowed us to confirm the presence of diverse iron species, verifying the compositional results from the EDS.

To determine the crystallographic forms of the iron oxides in the NIM, the X-ray diffraction (XRD) pattern reported was considered. Figure 1c reveals the existence of two main phases: one corresponds to α-Fe_2_O_3_ (hematite, which is predominant) and the other to the FeCO_3_ (siderite) structure [73,74]. These results are consistent with the information provided by the EDS and XPS and additionally suggest that Fe^3+^ ions are arranged in octahedral coordination with oxygen atoms, which are in a hexagonal close packing arrangement, forming a typical crystalline lattice of hematite [75]. Meanwhile, siderite typically crystallizes in the trigonal crystal system and exhibits a rhombohedral shape with curved and striated faces [76]. The rhombohedral siderite unit cell consists of CO_3_^2−^ anions and Fe^2+^ cations; the central Fe^2+^ cation is six-coordinated with a distorted (FeO_6_) octahedral geometry [77].

The semiconductor properties of the NIM were also studied. Diffuse reflectance spectroscopy (DRS) analyses revealed an absorption spectrum from 300 nm to 600 nm, showing signals corresponding to the intrinsic bandgap transition (Figure 1d, inset). Based on Kubelka–Munk’s estimation theory, the value of the bandgap energy was indirectly determined, being slightly greater than 2.1 eV (as shown in Figure 1d). This value belonged to the typical bandgap range reported for hematite in the literature (1.9–2.2 eV) [78,79]. The analyses of the surface of the material allowed us to determine, on one hand, the point of zero charge (PZC) of the material at 6.0, in Appendix A, which is useful for interpreting the results in terms of adsorption capacity, and on the other hand, the morphological characteristics of the iron mineral were observed using scanning electron microscopy (SEM). The corresponding micrographs are shown in Figure 1e and Appendix A, which demonstrate that the NIM has a grain size ranging from 0.04 to 1.0 μm. 

**Figure 1 molecules-29-05982-f001:**
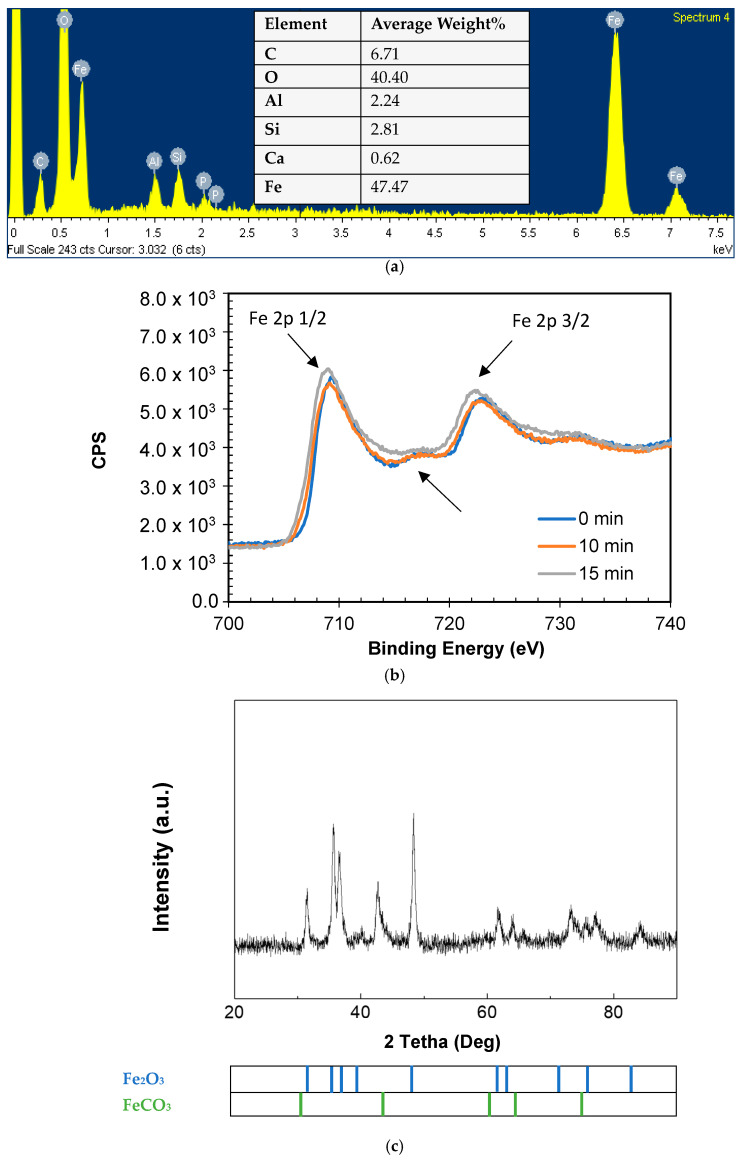
Characterization of NIM. (**a**) Energy-dispersive X-ray (EDX) analysis spectrum. (**b**) XPS, as received, after 10 min 500eV Ar+ and 5 min 5K eV Ar+ or without etching 500eV Ar+. (**c**) XDR (X-ray diffraction) for NIM sample. Solid line is the best fit of spectra [74]. (**d**) Optical diffuse reflectance spectra (DRS) (inset) and plot of (αhʋ)^1/2^ versus hʋ. (**e**) SEM analyses at different levels of magnification.

As seen in higher magnification levels, larger particles are agglomerates of semicircular nanoparticles (this fits well with the spherical morphology reported for hematite-based materials) [79]. The SEM micrographs (Figure 1e) also show that the NIM has a rough outer layer and a porous structure. Also, it is important to mention that the specific surface area of this iron mineral was reported in a previous work of our research team as 19.8 m^2^ g^−1^ [80]. According to recent reports, natural iron materials can facilitate photocatalytic reactions [49,81,82]. These reactions can take place through the semiconductor catalytic behavior of the material and/or their active role in the decomposition of hydrogen peroxide under conditions close to neutrality ((photo)Fenton-like reactions facilitated by the Fe^3+^ species on the surface of such materials) [46,83,84,85,86]. Interestingly, natural iron materials with a narrow bandgap can be utilized to generate charge carriers using solar light [49,54,80]. Hence, considering these distinctive properties, the NIM underwent photocatalytic tests to deal with the antibiotic ENR in aqueous media.

### 2.2. Photocatalytic Capacity of NIM for Enrofloxacin Degradation and Insights into Degradation Pathways

The capacity of the NIM to act as a photocatalyst in two systems (i.e., NIM–Light and NIM–Light–H_2_O_2_) to degrade ENR in water was initially evaluated (Figure 2a). As can be seen in Figure 2a, more than 85% of ENR was degraded after 180 min of treatment by the NIM–Light process, and the total removal of the antibiotic was achieved by applying the NIM–Light–H_2_O_2_ system for 120 min. Then, to obtain a comprehension of the causes of ENR elimination under these two photocatalytic systems, the eliminating action of the corresponding individual (i.e., NIM, Light, and H_2_O_2_; Figure 2b) and dual (H_2_O_2_–NIM and H_2_O_2_–Light, Figure 2c) subsystems was also assessed. The contribution of light to ENR degradation was determined. As can be seen in Figure 2b, when the ENR solution was exposed to sunlight solely, a decrease of up to 65% at 180 min of treatment was observed. In the same way, previous reports have indicated that the treatment using UV light results in only partial removal, highlighting the necessity of adding a photocatalyst to the system to enhance the elimination of these fluoroquinolones (FQs) [53,87]. The chemical structure of ENR, with carbonyl groups, heteroatoms, and conjugated pi-systems, allows it to absorb the ultraviolet components of solar radiation [88]. Indeed, the interaction of some fluoroquinolones (FQs) with solar light leads to direct and self-sensitized photo-transformations. According to Albini et al., the main pathway of direct photo-transformation is the cleavage of the C–F bond on FQs, Equation (1) [89] (more details about photo-transformations are presented below in Section 2.3). Nonetheless, the results in Figure 2b reveal the persistence of residual ENR even after 180 min of treatment. Therefore, only the photodegradation of ENR is not enough, and more elaborate systems (i.e., NIM–Light or NIM–Light–H_2_O_2_) are needed.
ENR + solar light → transformation products(1)

H_2_O_2_ + UVB → 2 HO^●^(2)

When ENR was exposed to H_2_O_2_, the process showed a limited degrading action (only ~10% was degraded after 180 min), indicating a low contribution of direct oxidation by hydrogen peroxide to the photocatalytic system. This is due to the fact that hydrogen peroxide is a poor oxidizing agent for ENR degradation [90]. However, the combined exposure of ENR to solar light and hydrogen peroxide at 180 min resulted in approximately 80% of FQ degradation, which was caused by the attacks of HO^●^. The hydroxyl radicals come from the homolysis of H_2_O_2_ promoted by the UVB component of sunlight radiation (300–320 nm) (Equation (2)) [90]. These results indicate the contribution of the H_2_O_2_–Light subsystem to the NIM–Light–H_2_O_2_ process. In addition to the antibiotic degradation caused by light or H_2_O_2_ acting individually or in combination, the interaction of the NIM with ENR under dark conditions was also considered (Figure 2b). The NIM exhibited a limited capacity for the removal of ENR, showing only 20% concentration reduction. This decrease in ENR levels could be attributed to the small adsorption of fluoroquinolone on the NIM. This result makes sense based on, on one hand, the low surface area (~20 m^2^ g^−1^) of the NIM and, on the other hand, the experimental pH of the solution (pH: 6.5). According to the point of zero charges of the NIM (PZC: 6.0, presented in the Appendix A), under experimental conditions, the surface of the NIM is predominantly negative. Meanwhile, at the experimental pH, ENR is a zwitterion due to the ENR structure having acidic and basic groups (pKa_1_ ~ 6.2, which is linked to the ionization of the carboxylic acid group, and pKa_2_ ~7.6, which corresponds to the release of the acidic proton on the piperazine ring) [91,92]. Therefore, under the experimental pH conditions, ENR may exhibit limited attraction to the charged NIM surface. Despite the zwitterion form of ENR having a positive charge, it also possesses another negative moiety, leading to electrostatic repulsion with the NIM surface charged negatively, thereby restricting the adsorption of FQ on the NIM. These results are consistent with another previous study on adsorption onto hematite- and siderite-based materials that also reports on the relevance of experimental pH and PZC for adsorption, remarking the possibility of higher adsorption levels under the conditions of pH > PZC for cationic species or pH < PZC for anionic or neutral species [93,94,95,96].

On the other hand, the simultaneous application of the material and hydrogen peroxide (i.e., H_2_O_2_-NIM control experiment) achieved an ENR removal not higher than 30% (Figure 2c). This removal was linked to the additive effect of each component (NIM and H_2_O_2_) on the target fluoroquinolone. Based on the above-presented results, there are contributions of ENR photolysis and adsorption on the solid material in the NIM–Light process. Moreover, from the interaction between the NIM and solar light, photocatalytic behavior is expected. Such a result can be attributed to the photo-activity of hematite within the NIM (see the material characterization in Section 2.1), acting as a semiconductor. The existence of a separation between the valence band (VB) and conduction band (CB) is evidenced in Figure 1d (indeed, the NIM has a bandgap of 2.1 eV). This allows irradiation to promote electrons from the VB to the CB, creating hole–electron pairs (Equation (3)). Subsequently, the electrons and holes promote the production of reactive oxygen species (ROS) (Equations (4) and (5)). It can be mentioned that the holes (Equation (6)) and ROS (Equation (7)) can oxidize organic compounds, such as ENR [84]. In line with these results, previous reports about the capacity of iron oxides, including hematite, suggested their ability to inactivate bacteria through photocatalytic processes [84,85]. Additionally, thanks to its photocatalytic behavior, hematite in aquatic settings can form complex compounds and promote the photoreduction of superficial ferric ions (Equations (8) and (9)) [97,98].
Fe_X_O_Y_ + solar light → Fe_X_O_Y_ (e^−^ + h^+^)(3)

e^−^ (CB) + O_2_ → O_2_^•−^(4)

h^+^ (VB) + O_2_^•−^ → ^1^O_2_(5)

h^+^ (VB) + FQ → FQ^•+^
(6)

^1^O_2_, O_2_^•−^ + FQ → transformation products(7)

>Fe^3+^- (L) + O_2_ + hv → Fe^2+^ + ROS(8)

>Fe^3+^-OH_(s)_ + hv → Fe^2+^_(aq)_ + HO^●^(9)

For the NIM–Light–H_2_O_2_ system (which exhibited the most effective outcome in degrading ENR, Figure 2a), in addition to the pathways present in NIM–Light, the homolysis of H_2_O_2_ has a high contribution. Also, the reduction of Fe^3+^ to Fe^2+^ by the reaction of the NIM with hydrogen peroxide joined in the formation of hydroxyl radicals (initiating a Fenton-like process; Equations (10) and (11)) is possible [5,37]. Indeed, several studies have reported that iron oxides (such as the hematite present in the NIM, according to the analyses presented in Section 2.1) contribute to water remediation [52,53,60,61,99,100]. Among others, Zhang et al. (2019) recently employed β-Fe_2_O_3_ with a narrower bandgap (1.9 eV), demonstrating its efficiency as a catalyst for photo-Fenton reactions in the degradation of pharmaceutical compounds due to its broader light response range [101]. Meanwhile, Liu et al. (2020) and Xiao et al. (2018) demonstrated that the enhanced visible light absorption of α-Fe_2_O_3_ also improved the photo-Fenton degradation performance of ciprofloxacin and acid red G (ARG) [102,103]. A detailed discussion of the role of iron species as a photocatalyst is presented in Section 2.3. Therefore, the high degradation efficiency toward ENR observed for the Light–NIM–H_2_O_2_ system occurs via the convergence of various mechanisms such as (i) photolysis, (ii) H_2_O_2_ homolysis, (iii) semiconductor action, and (iv) the heterogeneous photo-Fenton process. The photocatalytic action of the NIM for degrading ENR is supported by the activation of the solid material by solar light, as denoted by the characterization in Figure 1d. Furthermore, the participation of a heterogeneous photo-Fenton process is consistent with the presence of ferric and ferrous iron species at the NIM surface (as demonstrated by the XPS profile, Figure 1b). It is important to note that the NIM is composed of various minerals, so, to better clarify the role of each component, future work could be performed, testing the individual minerals in photo-treatments.
>Fe^3+^ + H_2_O_2_ → Fe^2+^ + H^+^ + HO_2_^●^(10)

>Fe^2+^ + H_2_O_2_ → Fe^3+^ + OH^−^ + HO^●^(11)

After recognizing the potentiality of the photocatalytic systems to degrade ENR, the ability of such processes to decrease antimicrobial activity (AA) was evaluated. AA is a relevant parameter that should be used during the treatment of antibiotics. Particularly, in the case of ENR, AA’s presence in water represents a contribution to the development of fluoroquinolone-resistant bacteria [104]. Thus, an evaluation of AA evolution against *S. aureus* during ENR degradation was performed, and it showed that the NIM–Light–H_2_O_2_ system completely decreased activity, while the NIM–Light process led to an AA decrease of ~70% (Figure 3). The understanding of ENR degradation and the decrease in its associated AA by the action of the photocatalytic processes demands both the elucidation of the involved ROS and the primary transformations of the target pharmaceutical. These two topics are developed in the subsequent sections.

### 2.3. Unraveling the ROS Involved in Water Remediation by Photolytic and Catalytic Systems

-
*Photolytic pathways*


As illustrated in Figure 2b, exposure to solar light resulted in a key starting point to allow for the degradation of ENR. Reports on the photochemistry of fluoroquinolones (FQs) suggest that their interaction with solar light can promote their transformation [105,106]. Initially, the ground state of FQs is excited to the lowest singlet state (^1^FQs*) (Equation (12)). This singlet state then undergoes intersystem crossing to the lowest triplet state (^3^FQ*) (Equation (13)). The triplet state (3FQ*) can evolve into defluorinated, dealkylated, and oxidized piperazine degradation products as a primary mechanism of photolytic degradation [107] (Equation (14)), or it can interact with dissolved oxygen through two pathways: (i) energy transfer, in which ^3^FQ* transfers energy to ground-state oxygen (^3^O_2_), producing singlet oxygen (^1^O_2_) (Equation (15)), and (ii) electron transfer, in which ^3^FQ* transfers an electron to ^3^O_2_, yielding a superoxide radical anion (O_2_^•−^) and a cation radical (FQ^+●^) (Equation (16)). The interaction of FQs with O_2_^•−^ and ^1^O_2_ is responsible for their transformation via self-sensitized pathways (Equation (17)). Some previous researchers also suggest that during the photodegradation of FQs, O_2_^•−^ can be converted into HO^•^ and promote FQ oxidation [108,109,110].
^1^FQ + solar light → ^1^FQ*(12)

^1^FQ*→ ^3^FQ*(13)

^3^FQ* → photoproducts(14)

^3^FQ* + ^3^O_2_ → ^1^FQ + ^1^O_2_(15)

^3^FQ* + ^3^O_2_ → FQ^+●^ + O_2_^●−^
(16)

O_2_^● −^ and/or ^1^O_2_ + ^1^FQ → degradation products(17)

To confirm the role of O_2_^•−^ and ^1^O_2_, aired solutions of ENR in the presence of scavengers were created. Para-benzoquinone (p-BQ) was used for O_2_^●−^ trapping (k_O2●_^−^_/p-BQ_ ~ 1.0 × 10^9^ L mol^−1^ s^−1^) [111]; furfuryl alcohol (AFF) was used to quench singlet oxygen (k_1O2/AFF_ = 1.0–1.2 × 10^8^ L mol^−1^ s^−1^); and the participation of holes was determined using potassium iodide (KI) [112]. As can be seen in Figure 4a, the degradation rates demonstrate the following order: without additives ≈ KI *>* AFF *>* p-BQ. This order suggests the participation of O_2_^●−^ > ^1^O_2_ >> h^+^ during ENR photodegradation. It should be mentioned that at a pH above 4.8, O_2_^●−^ could be transformed in its conjugate base, a hydroperoxyl radical (HO_2_^●^) (Equation (18)), which can evolve to H_2_O_2_ (Equations (19) and (20)) [113,114,115]. Thus, H_2_O_2_ presence was confirmed (Figure 4b). Although the degrading impact of hydrogen peroxide on ENR was minimal, as shown in Figure 2c, the UVB energy from sunlight (Equation (2)) and radical reactions could potentially facilitate the generation of HO^●^ (Equations (21) and (22)). A test with an isopropanol (IPA) scavenger for HO^●^ (IPA, k _HO●/IPA_ = 1.9 × 10^9^ L mol^−1^ s^−1^) was also performed, and as can be seen in Figure 4a, the results suggest its participation in ENR degradation [116].
HO_2_^●^ → O_2_^●−^ + H^+^
(18)

HO_2_^●^ + O_2_^●−^→ H_2_O_2_ + O_2_ k = 9.7 × 10^7^ M^−1^ s^−1^(19)

HO_2_• + HO_2_• → H_2_O_2_ + O_2_ k = 8.3 × 10^5^ M^−1^ s^−1^(20)

HO_2_^●^ + H_2_O_2_ → HO^●^ + O_2_ + H_2_O(21)

^3^ENR* + H_2_O_2_ → HO^●^ + O_2_ + H_2_O(22)

The degradation pathway results obtained with the scavenger tests were confirmed using radical-trapping experiments combined with electron paramagnetic resonance (EPR) spectroscopy (Figure 4c–e. DMPO and TEMP (see abbreviation definitions in Section 3) were used as spin traps of oxidizing species generated in the evaluated processes. DMPO was used to trap hydroxyl and superoxide radicals, and TEMP was used to trap singlet oxygen. The spin traps were independently added to ENR solutions in both the absence and presence of light. The signals of the adducts corresponding to each radical reactive oxygen species were elucidated considering the reported hyperfine one for each adduct [117,118,119]. As seen in Figure 4c, four characteristic signals of the DMPO-^●^OH adduct (marked with black symbols) can be distinguished during the photo-exposition of ENR, unlike the signals recorded in the control experiment, which was conducted in the absence of ENR (blue line). These results suggest the capacity of the self-sensitized photodegradation of ENR to promote the formation of HO^•^ radicals. HO^●^ presence was detected during the photodegradation of other fluoroquinolones such as ciprofloxacin (CIP) and Norfloxacin (NOR) [108,109,110]. However, according to the authors’ knowledge, the pathway that allows for their formation has not been reported. As mentioned before, HO^●^ generation would be possible as a consequence of the photoproduction of O_2_^●−^ (Equations (18)–(22)).

New measurements using EPR were performed to confirm the role of O_2_^●−^. To conduct these new measurements, it was considered that DMPO, besides trapping HO^●^, can also act as a spin trap for O_2_^●−^ and HO_2_^●^, forming the DMPO- O_2_^●^ and DMPO-^●^OOH adducts, respectively. However, their detection in the presence of HO^●^ in water is a challenge since (i) the reaction rates of DMPO with O_2_^●−^ (*k* = 2.0 − 170 M^−1^ s^−1^) or HO_2_^●^ (*k* = 6.6 × 10^3^ M^−1^ s^−1^) are considerably lower than those of DMPO with HO^●^ (*k* = 1.9–3.3 × 10^9^ M^−1^ s^−1^), and (ii) the half-life of DMPO-^●^OOH or DMPO-O_2_• does not exceed 0.8–1.3 min and evolves into DMPO-^●^OH, (k = 4.9 × 10^6^ M^−1^ s^−1^) [119,120,121,122,123], as shown in Appendix A. In this way, EPR measures were derived to eliminate the possible false positive signals of DMPO-^●^OH generated by DMPO-O_2_• or DMPO-^●^OOH evolution and to confirm O_2_^●−^ generation, using convenient conditions. The measurements for O_2_^●−^ detection were conducted in a mix of water–methanol (1:9), considering the selective scavenging capacity of methanol over HO^●^ [118]. As is observed in Figure 4c, the absence of DMPO-^●^OH signals (green line) confirm its generation during the photodegradation of ENR. In addition, measurements for determining O_2_•^−^ presence were performed using DMSO as a solvent. DMSO stabilizes O_2_•^−^, avoiding HO_2_^●^ formation. DMSO also acts as a scavenger of hydroxyl radicals (k = 7.0 × 10^9^ M^−1^ s^−1^), preventing its interference [122,124]. In Figure 4d, it is possible to identify the characteristic peaks of DMPO-^●^O_2_, and the increase in its intensities according to the time of irradiation of ENR becomes larger, and conforming to ENR, concentration is raised. These findings confirm O_2_^●−^ generation under the conditions used, which is consistent with previous reports on the use of iron oxides in water treatment [125]. In turn, Figure 4e shows the EPR spectrum using TEMP, and the characteristic triplets of the TEMP-^1^O_2_ adduct did not increase in signals once the system was irradiated in the presence of the molecule. These results are consistent with the minimal interference during scavenger tests using AFF. According to previous reports, ^1^O_2_ formation would be determined by the availability of ^3^ENR*, the precursor of ^1^O_2_, since ^3^ENR* can also give rise to photoproducts itself and may react with triplet oxygen to form a superoxide radical or ^1^O_2_ [69].

-
*Photocatalytic pathways*


To determine the ROS involved in the system based on the combination of the NIM with light (NIM–Light), scavengers were used initially. Also, EPR analyses were carried out. As seen in Figure 5a, the degradation rates in the presence of additives followed the following order: no scavengers > KI > AFF >> IPA >> p-BQ. These results suggest that the hole and singlet oxygen had minor participation in the NIM–Light system, whereas HO^●^ and O_2_^●−^ were strongly involved in this system. Such results were confirmed by the EPR analysis using DMPO and TEMP as spin traps.

The EPR results (Figure 5b) show the characteristic signals of DMPO-^●^O_2_, confirming superoxide radical anion formation, which is produced by oxygen reduction in a CB’s involvement in ENR degradation (Equation (4)) and sensitized photodegradation (Equation (16)). Crucial contributions by e− and h+ were also confirmed using scavengers during ofloxacin degradation without the involvement of O_2_^●−^ [111]. The absence of DMPO-^●^OH signals in Figure 5c denotes that there was no generation of hydroxyl radicals from the interaction of the NIM with light. However, in the presence of IPA, interference in ENR degradation during the NIM–Light process may occur due to the trapping of hydroxyl radicals generated from the interaction of H_2_O_2_ formed during ENR photolysis with solar light (Equations (16), (18)–(20)).

The EPR analysis using different concentrations of H_2_O_2_ demonstrates the viability of ENR degradation through the generation of H_2_O_2_ (Appendix A) and the formation of hydroxyl radicals via its photolysis and/or the reaction of H_2_O_2_ with Fe²⁺ present in the NIM. It is important to mention that in the NIM–Light system, a small amount of singlet oxygen is found, as supported by the EPR analyses using TEMP (Figure 5d). The typical triplet signal for the TEMP-^1^O_2_ adduct is observed. Singlet oxygen can be formed by the interaction of light with ENR and the reaction of O_2_^●−^ with the photogenerated hole in the NIM (Equations (5) and (15)).

Concerning the NIM–Light–H_2_O_2_ system, the experiments using scavengers (Figure 6a) showed the following degradation order for ENR: no scavenger ≈ AFF ≫ p-BQ > IPA. Additionally, the EPR measurements conducted with DMPO in water or DMSO revealed the characteristic signals of DMPO-^●^OH and DMPO-^●^O_2_, respectively (Figure 6b,c). The possible leaching of iron ions from the NIM was demonstrated by finding concentrations in the order of ppb (Appendix A) which could contribute to a homogeneous photo-Fenton process. As explained above, the superoxide anion radical comes from the interaction of dissolved oxygen with the electrons in the CB of the NIM. In turn, the hydroxyl radical is formed by H_2_O_2_ homolysis (Equation (2)) and (photo) Fenton-based processes (Equations (23)–(29)) with the participation of iron in heterogeneous (i.e., NIM) and homogeneous (from NIM leaching) forms, as reported by Mohammadi, S. et. al., (2021), who extensively describe the degradation pathways in remediation processes promoted by iron oxides [126].
>Fe^3+^- OH + H_2_O_2_ → (H_2_O_2_)_s_(23)

(H_2_O_2_)_s_ → (>Fe^2+^ HO_2_^●^) + H_2_O(24)

(>Fe^2+^-HO_2_^●^) → >Fe^2+^ + HO_2_^●^(25)

>Fe^2+^ + O_2_ → >Fe^3+^-OH + HO_2_^●^(26)

>Fe^2+^ + H_2_O_2_ → >Fe ^3+^-OH + HO^●^ + H_2_O(27)

Fe^2+^ + H_2_O_2_ → Fe^3+^ + OH^−^ + HO^•^(28)

Fe^3+^ + H_2_O_2_ → Fe^2+^ +HO_2_^•^ + H^+^(29)

### 2.4. Connection Between Photocatalytic Elimination of Antibiotic Activity (AA) with Primary Degradation Products

#### 2.4.1. AA Removal Considering Two Representative Microorganisms

As mentioned before, AA is a critical aspect to consider during the degradation of antibiotics in water. Analyzing AA allows us to determine the capacity of the processes to eliminate or reduce the risks associated with drug resistance generation. Herein, two previous reported representative microorganisms, *S. aureus* (Gram-positive bacteria) and *E. coli* (Gram-negative bacteria), were used to account for AA evolution [127,128,129,130,131]. A comparison between the AA decrease by treatments and standard solutions was also considered (Figure 7). As reflected in Figure 7a,b, both bacteria exhibit susceptibility to the evaluated standard concentrations greater than 0.6 mg/L for *E. coli* and 1.0 mg/L for *S. aureus*, their minimum inhibitory concentration (MIC) under the used conditions for the AA analyses.

Figure 7a also shows that both photocatalytic processes (i.e., NIM–Light and NIM–Light–H_2_O_2_) were able to eliminate AA against *S. aureus.* However, the solutions obtained during the application of the NIM–Light and NIM–Light–H_2_O_2_ processes caused more growth inhibition for *E. coli* than the ENR standards, indicating that in the case of the Gram-negative bacteria, AA remained even when fluoroquinolone was 100% degraded (Figure 7b). Such differences between the two representative microorganisms may be related to both the interaction of the generated products and the characteristics of each bacterium type.

The effectiveness of an antibiotic is influenced by its chemical composition and the way it interacts with the targeted microorganism. Some authors report the distinct action of FQs on essential bacterial enzymes such as DNA gyrase and DNA topoisomerase IV. Both enzymes work together in DNA replication, transcription, recombination, and repair. In Gram-negative bacteria, gyrase is more susceptible to inhibition by quinolones than topoisomerase IV, whereas in Gram-positive bacteria, topoisomerase IV is often the primary target, and gyrase is inherently less susceptible. These differences may be responsible for the distinctive response to ENR [132]. In this regard, it is of interest to determine the origin of the remaining AAs for *E. coli*. Therefore, in the following section, the structure of the generated products and the analysis of their possible AAs are discussed.

#### 2.4.2. Structure of Primary Transformation Products 

The stable primary transformations of the antibiotic under the photocatalytic treatments of ENR were detected. The chromatograms at approximately 75% of ENR degradation were analyzed (Figure 8). Signals at 6.8 min corresponded to the retention time (RT) for ENR, and signals with lower RTs represented the transformation products generated in each system. The exposure of ENR to NIM–Light generated five stable initial transformation products with RTs of 2.9 min (P1), 3.1 min (P2), 4.2 min (P3), 4.9 min (P4), and 5.2 min (P5). Meanwhile, the chromatogram for NIM–Light–H_2_O_2_ showed extra signals that corresponded to five additional degradation products, with RTs of 1.9 min (P6), 2.6 min (P7), 3.2 min (P8), 4.2 min (P9), 5.1 min (P10), and 6.3 min (P11). To elucidate the chemical structure of such primary degradation products, LC-MS analyses were conducted, and some reactions involved in their formation were proposed. Appendix A summarizes some characteristics of these products, and Figure 9 schematizes the possible initial degradation pathways. Some of these products (specifically P1, P2, P3, P4, and P5) have also been reported during the degradation of ENR under the photocatalytic treatment using wüstite [63]. However, NIM-based systems led to the formation of other new intermediates, which could be explained by a possible higher production of ROS in the NIM–Light–H_2_O_2_ process.

The generation of P1 occurs as a consequence of the photo-substitution mechanism promoted by light, in which the fluorine atom is replaced by a hydroxyl moiety [89,105,106]. P1 can experience the addition of a hydroxyl group to form P5. The formation of P3 is the result of ROS action on the piperazyl ring, leading to its cleavage. The subsequent defluorination of P3 [106] generates P2, also found in [89,106]. P4 is the result of the cleavage of ethyl group rupture attached to the piperazyl ring, reported also by [110,133,134]. Additionally, the hydroxylation of the ethyl chain linked to piperazyl results in the formation of P10. This last hydroxylation is likely accompanied by oxidations in the piperazyl group, which may facilitate its cleavage to form P6. Subsequently, P6 undergoes a rupture of the ethyl group attached to the piperazyl ring, leading to the generation of P7 and P8. We should mention that P7 is formed as a consequence of the double bond cleavage of the quinolone nucleus plus subsequent hydroxylation, while in P8, hydroxylation occurs in one of the carbons of the remnants of the piperazyl ring. Finally, it can be noted that P9 comes from the piperazyl ring cleavage without the loss of additional atoms.

#### 2.4.3. Transformation Products and Their Link with AA Removal

In our previous study using wüstite, it was shown that the AA of some ENR photoproducts, P1, P4, and P5 in particular, exhibited the inhibition of *E. coli* growth but not against *S. aureus* [63]. In the current paper, to discuss the origin of the remaining AA removal (Figure 7), a qualitative approach was used. The inhibitory effects of five extra transformation products obtained during the NIM–Light–H_2_O_2_ process on the growth of *E. coli* were tested. To perform the AA analyses, an ENR solution, five times more concentrated than the previous solutions, underwent exposure to the NIM–Light–H_2_O_2_ process for 120 min, reaching approximately 75% ENR degradation. The resulting products were isolated by collecting them based on their retention times during HPLC analysis. Subsequently, the presence or absence of inhibition zones in agar diffusion tests was studied, and it was found that P9 and P10 inhibited the growth of *E. coli*.

At this point, it is important to remember that fluoroquinolones lead to bacterial death by inhibiting IV topoisomerase and DNA gyrase, enzymes vital for modulating the chromosomal supercoiling necessary for DNA synthesis, transcription, and cell division [132,135,136,137]. The interaction between bacteria and fluoroquinolones occurs by the formation of ternary complexes FQ-enzyme–DNA, facilitated by the drug skeleton’s 3-oxo-4-carboxylic acid structure. Those complexes inhibit the supercoiling activity of the DNA gyrase enzyme, thus exerting their antibacterial action on DNA and RNA synthesis, resulting in a biphasic response and the killing of susceptible organisms [132,135,136,138].

Considering the structural factors of fluoroquinolones for AA, in addition to the carboxylic acid group, there are another two key moieties: the piperazyl ring and the fluorine atom. The piperazyl ring aids in enhancing efflux inhibition penetration, preventing drug expulsion from bacterial cells, and prolonging its activity. In turn, the fluorine atom improves cell permeability and impedes DNA replication within bacteria, further disrupting their ability to proliferate [135,139,140,141]. Thereby, the response of *E. coli* to P9 and P10 makes sense because they could interfere with the cell division cycle thanks to the presence of carboxylic groups and fluorine atoms. Furthermore, P10 retains the piperazyl moiety, and although P9 does not have the piperazyl ring, it presents a part that still resembles such a moiety.

### 2.5. Treatment Extent: Testing of Photocatalytic Processes in Disinfection, Mineralization, and Reuse Cycles of NIM

After the evaluation of the photocatalytic systems for degrading ENR and decreasing AA, assessed their disinfecting ability was assessed. For such purpose, *E. coli* was considered as the target microorganism. Figure 10a presents the inactivation of this bacterium under the NIM–Light and NIM–Light–H_2_O_2_ systems. Also, the control experiments of light or the NIM acting alone were performed, and the results are included in Figure 10a. A small inactivation of *E. coli* (~0.5 log units in 60 min) by direct interaction with the NIM was found (which could be associated with a weak direct interaction of the microorganism with the mineral surface). At the experimental pH (> 6.0), the NIM is negatively charged, and the bacteria carry a negative charge externally. Moreover, the NIM has a grain size of less than 1 μm (see Figure 1e), while *E. coli* consists of μm sized microorganisms [126]. Hence, the adsorption on the NIM surface and the formation of aggregates is low; thus, the bacterial population is not affected significantly. Additionally, the UVA and UVB of solar light components have disinfecting action (they induce alterations of DNA or enzyme inactivation [125], explaining the *E. coli* population reduction (~2 log units after 60 min) by the light subsystem. As observed in Figure 10a, the inactivating capabilities of NIM–Light and NIM–Light–H_2_O_2_ were superior to those of the control tests. In fact, the NIM–Light–H_2_O_2_ process achieved a ~ 4.5-log reduction after 60 min of *E. coli* treatment. This was attributed to the higher production of ROS in the photocatalytic process. ROS such as HO^●^ are stronger disinfecting agents since the NIM–Light–H_2_O_2_ system was shown to be more efficient than NIM–Light for inactivating *E. coli* (Figure 10a), degrading ENR (Figure 2), and decreasing AA (Figure 7). Similar results were recently reported by De la Obra Jiménez, who explored the use of natural iron oxides (FeOx) as photo-Fenton catalysts to remove bacteria from secondary effluents using a bench-scale prototype. Their results showed that FeOx achieved a 5-log removal of fecal bacteria, comparable to the most effective Fe^2+^ salt-based photo-Fenton configuration [50]. Furthermore, the extent of the treatment for ENR mineralization by TOC control was tested in this research. Figure 10b shows the evolution of TOC, AA, and ENR concentration.

Mineralization can require more attacks by the ROS beyond the initial transformations of the target pollutant (which are responsible for the diminution of ENR concentration and AA decrease). Then, after 240 min of treatment, ~26% of TOC was removed, while AA and ENR were completely decreased at 180 min. As the hole (which has a high mineralizing capability) showed a low involvement in antibiotic degradation (Figure 3), the observed removal of TOC was explained by the consecutive responses of ENR or its primary degradation products (P1-P10, Figure 9) toward the degrading ROS attacks. Such consecutive attacks can induce several structural transformations up to the generation of carboxylic moieties (RCOOH/RCOO^−^). These last kinds of functional groups can be attacked by HO^●^ releasing carbon dioxide [142,143,144]. Also, the ferric ions in solution can be complexed by such carboxylic moieties and then experience photodecomposition, enhancing mineralization [54,98,99,145].

Another important topic in heterogeneous photocatalytic processes is the reuse of catalysts [146,147]. Thereby, in our case, the reusability of the catalyst in NIM–Light–H_2_O_2_ was examined, with the decrease in AA against *S. aureus* as the response variable (Figure 11). After each use, the iron oxide catalyst was recovered, and its photocatalytic activity was evaluated in three subsequent reuse cycles, under identical conditions of irradiation and H_2_O_2_ concentration. It was found that the good performance of the system (a complete AA decrease after 180 min of treatment) remained across these cycles. These findings are consistent with previous reports on targeting organic pollutants, which remark on the high reusability of natural iron oxides in photocatalytic processes [147].

To evaluate the capacity of the NIM–Light–H_2_O_2_ system in a real scenario, the treatment of ENR in an effluent of a municipal wastewater treatment plant (EWWTP) was carried out. To perform this experiment, the EWWTP was spiked with ENR at the same concentration used in distilled water (Appendix A shows the effluent composition and methods used to measure parameters like pH [148], conductivity [149], chemical oxygen demand (COD) [150], turbidity [151,152], total organic carbon (TOC) [153], and dissolved oxygen [152,153] The evolution of the associated AA followed. Figure 12 compares the decrease in antimicrobial activity in pure water and the EWWTP by the NIM–Light–H_2_O_2_ process.

It can be observed that the removal of AA in the real wastewater matrix occurred at a slower rate compared to pure water. Interestingly, the NIM–Light–H_2_O_2_ system was able to completely decrease the activity in the EWWTP after 240 min. Among the reasons that could affect the process performance in the real matrix, the screening effect of solar light caused by the matrix components (e.g., turbidity, Appendix A) could be relevant. Moreover, the presence of inorganic ions (e.g., chloride or sulfate, as indicated by sample conductivity) and organic matter compete with ENR for reactive oxygen species (ROS). Consequently, the AA decrease in the EWWTP is slower than that in pure water. These outcomes of treating the complex matrix using the NIM–Light–H_2_O_2_ system are similar (i.e., slower performance in complex water) to the findings reported for other processes involving the use of iron oxides as catalysts for the degradation of organic compounds, whether in artificial or real wastewater [50,53,100,154].

## 3. Materials and Methods

### 3.1. Reagents

Enrofloxacin, ENR (>98%, 0.05 mmol L^−1^), and formic acid (98–100%, 0.1 and 0.02% *v*/*v*) were provided by Sigma-Aldrich Chemie GmbH (Steinheim, Germany). HPLC-grade acetonitrile, ACN (≥99.9%, 15% *v*/*v*); isopropanol, IPA (99%, 0.1 mol L^−1^); hydrogen peroxide, H_2_O_2_ (30% *w*/*v*, 1 mmol L^−1^); methanol, MeOH (99.8%, 90% *v*/*v*); and NaCl (99.5%, 0.9%*w*/*v*) were acquired from Panreac AppliChem ITW and Probus reagents (Barcelona, Spain). Ammonium metavanadate, NH_4_VO_3_ (99%, 0.06 mol L^−1^); Ammonium acetate, CH_3_COONH_4_ (99%, 0.1 mol L^−1^); p-benzoquinone, p-BQN (>98%, 0.1 mol L^−1^); Ferrozine^®^ (98%. 0.01 mol L^−1^); ferric chloride, FeCl_3_ (99%, 0.018 mol L^−1^)hydrochloric acid, HCl (37%, 0.01 mol L^−1^); and hydroxylamine hydrochloride, NH_2_OH∙HCl (99%, 1.4 mol L^−1^), were purchased from Sharlau (Barcelona, Spain). Furfuryl alcohol, AFF (98.8%, 0.1 mol L^−1^) from Merck (Johannesburg, South Africa). 5,5-dimethyl-1-pyrroline N-oxide, DMPO (97.0%, 5.0 mmol L^−1^), and dimethyl sulfoxide (DMSO) (99.5%) were purchased from Cayman Chemical (Neratovice, Czech Republic). 2,2,6,6-tetramethylpiperidine, TEMP (98%, 5.0 mmol L^−1^), and Barium sulfate, BaSO₄ (97%), were provided by Thermo Scientific (Kandel, Germany).

Müller Hilton Agar, Nutritive Agar, Agar MacConkey, Agar Slanetz & Bartley, and TTC 1% (triphenyl tetrazolium chloride) were supplied by Scharlau, (Barcelona, Spain).

The natural iron material (NIM) came from an iron mine in the Andean Colombian region (Boyacá, Colombia). NIM particles were sieved using screens with pore sizes smaller than 75 microns.

### 3.2. Characterization of Natural Iron Material

The NIM was analyzed using various techniques. The morphology of all samples was analyzed using field scanning electron microscopy (SEM) with a JEOL JSM 6400 instrument, manufactured by JEOL Ltd (Tokyo, Japan). To analyze the chemical composition of the materials, X-ray Energy-Dispersive Spectrum (EDS) analyses were performed using an EDAX detector provided by Ametek Inc. (Wiesbaden, Germany). X-ray photoelectron spectroscopy (XPS) studies were conducted on the natural iron material using a Kratos AXIS Supra XPS spectrometer manufactured by AMETEK, Inc. (Manchester, United Kingdom) with monochromatic Al Kα X-rays at 225 W (15 kV, 5 mA). Survey scans were acquired with a pass energy of 160 eV and a step size of 1000 meV. Narrow scans were performed for the O 1s, Fe 2p regions, and the valence band, with a pass energy of 20 eV and a step size of 100 meV, with two sweeps acquired for each narrow scan region. Initially, the surface of the samples was analyzed without any modification or cleaning, examining three different zones to verify homogeneity. The analysis was conducted using the Hybrid slot lens mode, corresponding to an area of approximately 700 × 300 μm. The subsequent analysis was performed after 10 min of etching with 500 eV Ar+ and an additional treatment of 5 min at 5 keV Ar+. Finally, the light absorption properties of samples were studied using UV–vis spectroscopy. UV–vis diffuse reflectance (UV–vis DRS) spectra were recorded using a Varian Cary 100 spectrophotometer, developed by Varian, Inc., (Palo Alto, CA, USA) equipped with an integrating sphere, with BaSO₄ serving as the reference compound. The bandgap value was calculated using the Tauc method, by plotting (αhʋ)^1/2^ versus hʋ, considering the indirect transition characteristic of the semiconductor material.

### 3.3. Reaction Systems

All experiments were conducted on batch systems at the laboratory scale using Pyrex^®^ reactors of 500 mL (Châteauroux, France). Each test consists of an aqueous sample of 250 mL containing one of the selected pollutants in distilled water or a real aqueous sample. The addition of catalysts and/or H_2_O_2_ was carried out in a single dose, and the light source was activated to initiate the tests. The concentration of ENR was 0.05 mmol L^−1^, considering the LOQ for the HPLC methods utilized to follow the evolution during the treatments. The experiments were conducted at a natural pH (6.5) under constant stirring using a Dragon Lab MS-M-S10 magnetic stirrer (power 0.02 kW), manufactured by Dragon Laboratory Instruments (Beijing, China). Control experiments for each process were conducted using the same parameters but under dark conditions. Each experiment was performed at least in duplicate. All solutions were prepared with distilled water, except for those solutions used for chromatographic analysis, which were made with purified water from a Millipore Milli-Q^®^ system. The concentration of the antibiotics in real water was ensured by adding 25 mL of a 0.5 mmol L^−1^ solution of the contaminant to 225 mL of the real sample.

For all light-assisted experiments, sun radiation was provided by a solar simulator (Hanau Suntest) equipped with an air-cooled Xenon lamp, and appropriate filters were employed to achieve a cutoff at 300 nm manufactured by Atlas Material Testing Technology (Columbus, OH, USA). The illumination intensity for the UVA component was measured using an LS126C radiometer produced by Shenzhen Linshang Technology Co., Ltd., (Guangdong, China). The emitted radiation exhibited the following characteristics: 0.5% in the UVB range (300–320 nm) and 5–7% in the UVA range (320–400 nm). Above 400 nm the solar spectrum is fully simulated. The temperature inside the reactor never exceeded 35 °C.

Disinfection experiments were performed with *Escherichia coli* ATCC 25922. The strain used in this study, ATCC 25922, was obtained from the American Type Culture Collection (Manassas, VA, USA). Strains that grow in aerobic conditions were routinely streaked from frozen stocks to Nutritive Agar until its stationary phase was reached by cultivation at 37 °C. The bacterial colonies were suspended in distilled water, and the suspension concentration was controlled considering an optical density of 0.35 a λ = 625 nm, which generated a concentration of around 10^9^ CFU mL^−1^, from which test water was spiked to an initial concentration of 10^6^ CFU mL^−1^.

### 3.4. Analyses

*- Monitoring and Control of advanced processes:* The continuous data collection of pollutants, including antibiotic concentration, antibiotic activity, total organic carbon, and Colony-Forming Units (CFU), provides crucial insights into the efficiency of processes. These three fundamental parameters were monitored to assess the effectiveness of the applied methods. In addition, control measures, such as pH, the solved iron, electronic paramagnetic resonance for detecting radical species, and product identification through UPLC/mass spectrometry, among others (as is describe below) play a pivotal role in gaining an understanding of factors such as the degradation pathway, the types of oxidant species generated, and the extent of treatment.

*- Chromatographic Analyses:* The degradation of ENR in distilled water was assessed using high-performance liquid chromatography (HPLC). Samples (1.0 mL) were collected at specific time intervals for analysis. Fluoroquinolone concentration changes were monitored using a Waters 2695 HPLC instrument, equipped with a C-18 superxcel (100 × 4.6 mm 3 μm) reverse phase HPLC column produced by Waters Corporation (Milford, MA, USA). The optimal separation was carried out using a mixture of formic acid 0.02% and acetonitrile (85:15 *v*/*v*) as the mobile phase, at 30° C and under isocratic conditions. Quantification was carried out by UV detection at 280 nm. The flow rate was 1.0 mL min^−1^, and 10 μL was the injection volume, as described before [63].

The identification of transformation products was carried out by UPLC/MS analysis in a Waters Acquity instrument coupled to a Waters Acquity QDa single quadrupole mass spectrometer (Milford, MA, USA). The column used was a C-18 superxcel (100 × 4.6 mm 3 μm) reverse phase column, and the eluent was a mixture of water and acetonitrile; both solvents were doped with 0.1% formic acid, (85:15 *v*/*v*), the injection volume was 10 μL, and the temperature was 30° C. A detector was operated in positive ESI mode using a scan range of 130–1500 *m*/*z*.

*- Antibiotic Activity* (AA): The Kirby–Bauer method or agar diffusion method was used to determine antibacterial activity at different times of treatments or for antibiotic standard solutions according to a previous report [63]. For this analysis, *E. coli* (ATCC 25922) and/or *S. aureus* (ATCC 29213), obtained from the American Type Culture Collection (Manassas, VA, USA) were used as indicator microorganisms. The diameters of the inhibition halo were measured in mm to determine the decrease in antibiotic potency during the treatments [155].

*- Total Organic Carbon (TOC):* To evaluate the degree of mineralization, TOC was measured according to method 5310 B of the Standard Methods for the Examination of Water and Wastewater [156]. TOC was determined using a Shimadzu LCSH TOC analyzer (Kyoto, Japan) (standard error less than 2%).

*- Preparation of Escherichia coli Suspensions*: The microorganisms in the samples collected during the experiments were quantified using the standard plate counting method, involving serial 10-fold dilutions in sterile saline solution, as detailed recently [157]. If the bacterial concentration was below 10 CFU mL^−1^, the samples underwent the filtration method. A 10 mL sample was filtered using cellulose nitrate membranes with a pore size of 0.45 µm in a filtration ramp, produced in Darmstadt, Germany [157]. The detection limit (DL) of this technique is 10 CFU 100 mL^−1^, which is the minimum disinfection level required by the European regulation on water reuse for Class A treated municipal wastewater, EU 2020/741, adopted by the European Parliament and the Council of the European Union on 25 May 2020, and entered into force on 26 June 2020. It became applicable on 26 June 2023, providing the necessary framework and standards for the safe reuse of treated urban wastewater.

*- Concentration of Dissolved Iron:* This was followed by the Ferrozine method [158,159] using a Genesys spectrophotometer, manufactured by Thermo Fisher Scientific (Milwaukee, WI, USA) and absorbance measurement at 562 nm [63].

*- Hydrogen Peroxide Evolution:* This was determined by the ammonium metavanadate method [160]. An aliquot of 500 μL from the reactor was added to a quartz cell containing 1500 μL of NH_4_VO_3_ (0.06 mol L^−1^); then, the absorbance was measured using a Genesys spectrophotometer and absorbance measurement at 450 nm, and quantification was performed using the Beer law.

*- Counting of Escherichia coli:* The microorganisms present in the samples collected during the experiments were quantified using the standard plate counting method described above in the preparation of *E. coli* suspensions.

*- pH Evolution:* To determine the pH of the aqueous samples, a CRISON brand pH meter, model GLP 21 (Allela, Spain), was used. The method followed was 4500-HB from the Standard Methods. [153].

### 3.5. Analysis of Degradation Routes

During the degradation processes, several active radicals can form in media. To determine their participation, experiments using EPR spin traps were conducted. Additionally, the identification of principal degradation processes was analyzed through Mass-coupled ultra-performance liquid chromatography, as was described above. The identification of species using EPR analysis was performed using an X-band Bruker EleXsys E580 spectrometer equipped with an ELEXSYS Super High Sensitivity Probehead (Karlsruhe, Germany). The HO^●^, O_2_^●^, and SO_4_^●−^ trapping reactions were performed using 5,5-dimethyl-1-pyrroline N-oxide (DMPO). To trap ^1^O_2_, 2,2,6,6-tetramethylpiperidine (TEMP) was used according to [157]. The EPR instrument was operated at a microwave (mw) frequency of 9.84 GHz and a mw power of 19 mW. The magnetic field was modulated at 100 kHz, and the modulation amplitude was set to 0.1 mT after having checked that no overmodulation effects were observed at this modulation amplitude.

### 3.6. Cycles of Reuse

To test the reusability of the NIM, various consecutive reuse cycles were considered. At the end of each cycle, iron-based materials were collected by the filtration of the treated solution and then dried in an oven at 105 °C. Then, dried wüstite was added to a new solution of pollutant for the next reuse cycle.

## 4. Conclusions

The characterization of the NIM revealed that this material has a grain size at a micrometer range, featuring semicircular forms and a rough outer layer. Moreover, the NIM is composed of a mixture of Fe^+3^ and Fe^+2^ species, predominantly Fe_2_O_3_ (hematite) and to a lesser extent FeCO_3_ (siderite). Due to a low bandgap, the NIM acted as a semiconductor under solar light irradiation. The NIM-based photocatalytic systems were useful for degrading ENR and decreasing its associated AA. The integration of simulated solar light with the NIM diminished the concentration of ENR and the associated antimicrobial activity (AA) but did not achieve complete removal. However, the introduction of hydrogen peroxide into the system containing NIM and light facilitated the rapid generation of reactive oxygen species (ROS), thereby enhancing the elimination of ENR and AA. The EPR analysis confirmed the involvement of O_2_^●−^ in the degradation of ENR, with the NIM serving as a photocatalytic semiconductor. In turn, O_2_^●−^ and HO^●^ have a preponderant role in the NIM–Light–H_2_O_2_ process. The exposure of ENR to the NIM–Light system resulted in the formation of five degradation products, while the introduction of H_2_O_2_ into the system (NIM–Light–H_2_O_2_ process) generated five additional degradation products. The identified intermediates did not exhibit activity against *S. aureus*. However, some structural changes in the initial molecule were not enough to eliminate the AA against *E. coli* (specifically, five of the generated products were active against this Gram-negative microorganism), and such differentiated AA action was linked to the type of microorganism. The NIM–Light–H_2_O_2_ system demonstrated the best performance in reducing AA, even in a complex matrix such as the EWWTP, as well as reducing microbiological contamination by achieving reductions of 5 logarithmic units, which is reasonable considering the reactive oxygen species that were formed. Furthermore, the NIM can be reutilized for up to three cycles while maintaining 100% efficiency in decreasing AA. This research could be enriched in future work with analyses focused on better clarifying the role of each component, testing the individual minerals in the photo-treatments, increasing the volumes of the real residual water treated with and without the addition of the model contaminant, testing other compounds under similar conditions as ENR, and using natural solar radiation or LED lamps, in order to determine the technical and economic feasibility of the technology at a larger scale.

## Figures and Tables

**Figure 2 molecules-29-05982-f002:**
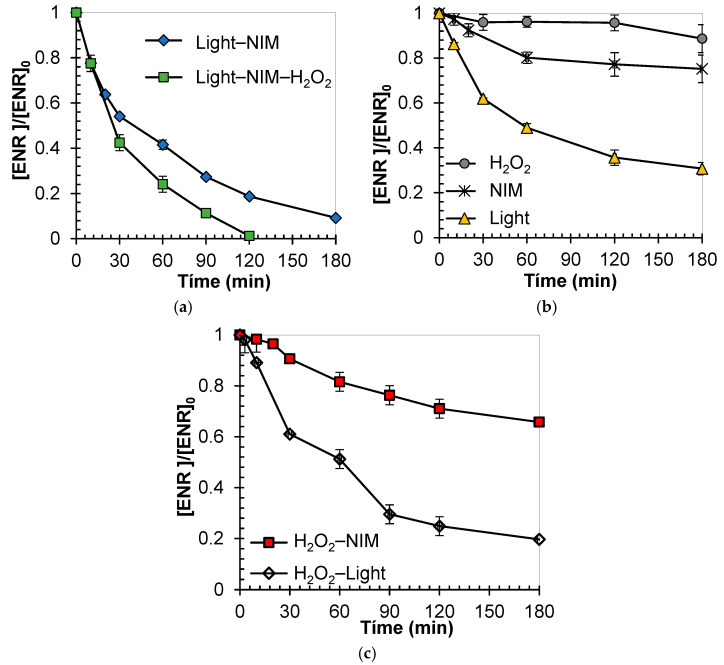
Evaluation of ENR transformation by (**a**) Light–NIM and Light–NIM–H_2_O_2_ processes, (**b**) individual processes (NIM, Light, and H_2_O_2_), and (**c**) dual processes (H_2_O_2_–Light and H_2_O_2_-NIM). Experimental conditions: [ENR] = 0.05 mmol L^−1^; [NIM] = 10.0 mg L^−1^; [H_2_O_2_] = 1.0 mmol L^−1^; pH = 6.5; for irradiated systems, light intensity = 500 W m^−2^.

**Figure 3 molecules-29-05982-f003:**
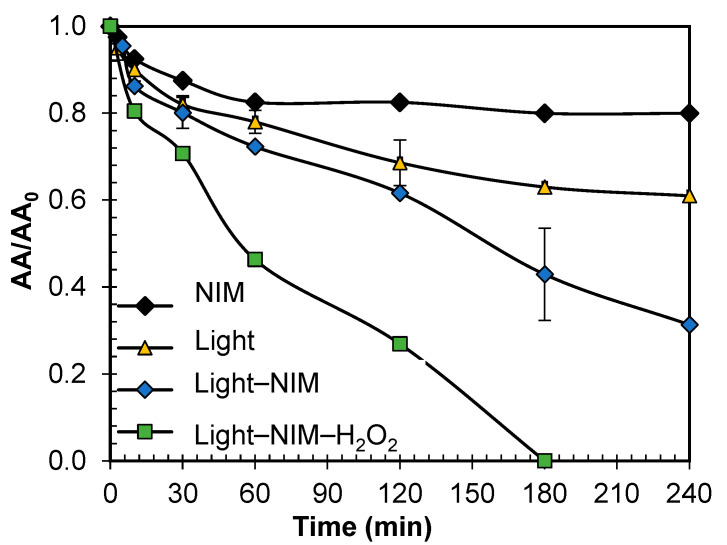
AA elimination associated with ENR under different subsystems using NIM. Experimental conditions: [ENR] = 0.05 mmol L^−1^; light intensity 500 W m^−2^; [NIM] = 10.0 mg L^−1^; [H_2_O_2_] = 1.0 mmol L^−1^; pH = 6.5.

**Figure 4 molecules-29-05982-f004:**
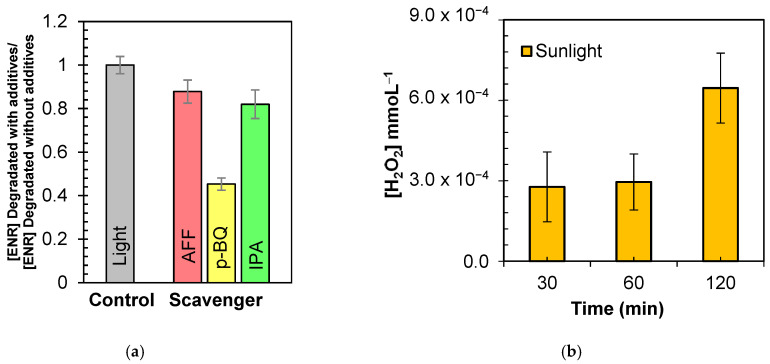
(**a**) ROS involved in photodegradation of ENR after 60 min of sunlight exposure in absence and presence of scavengers. (**b**) Accumulation of endogenous H_2_O_2_ during ENR degradation under sunlight. EPR spectrum on radicals captured under sunlight system using (**c**) DMPO in water and water–methanol (1:9 *v*/*v*), (**d**) DMPO in DMSO (under these conditions the effect of irradiation time and ENR concentration was evaluated)., and (**e**) TEMP in water.

**Figure 5 molecules-29-05982-f005:**
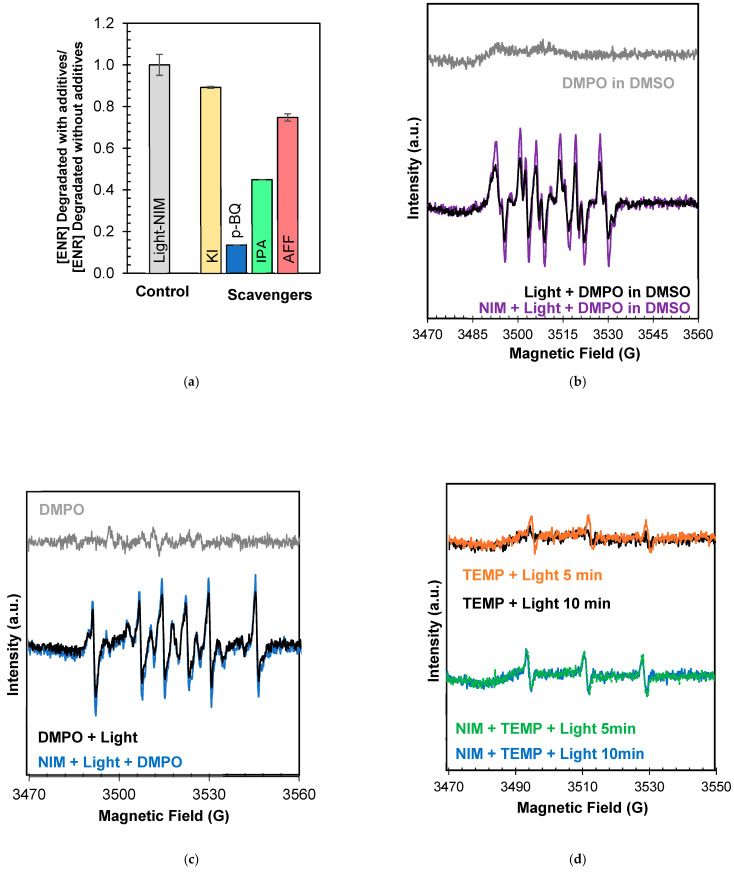
ROS involved in photodegradation of ENR by NIM–Light system. (**a**) ROS involved in photodegradation of ENR after 60 min of treatment in absence and presence of scavengers. EPR spectrum of radicals captured under NIM–Light system by (**b**) DMPO in DMSO, (**c**) DMPO in water, and (**d**) TEMP in water.

**Figure 6 molecules-29-05982-f006:**
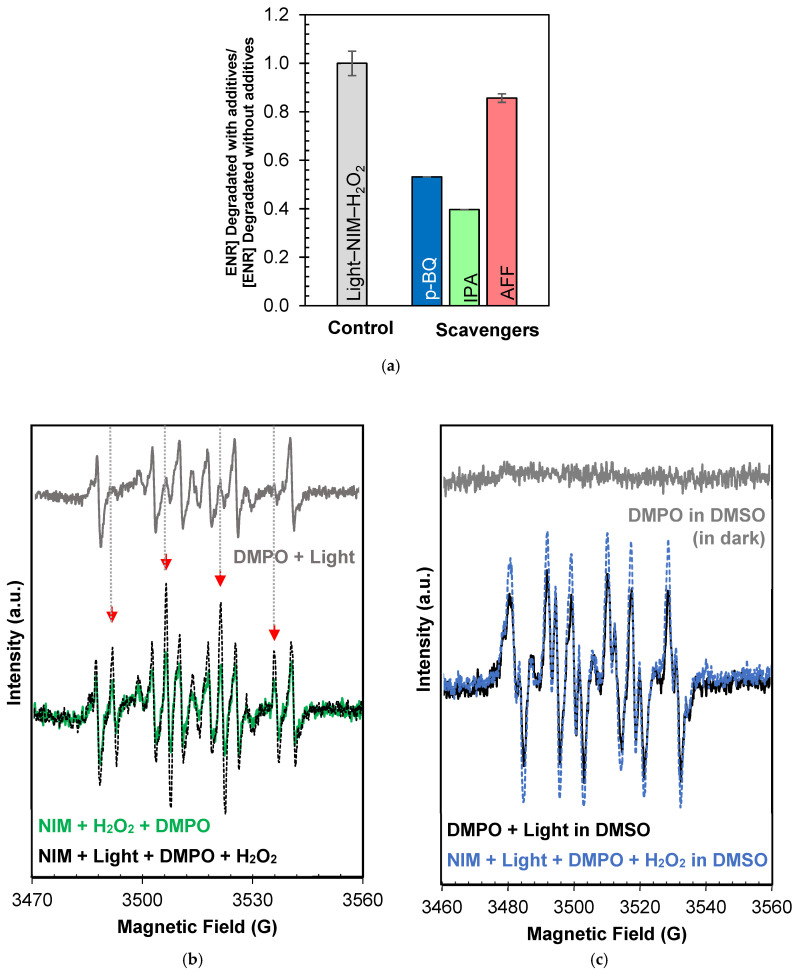
Identification of ROS involved in degradation of ENR by NIM–Light–H_2_O_2_ system. (**a**) Antibiotic treatment by NIM–Light–H_2_O_2_ system in absence and presence of scavengers. EPR spectrum of radicals captured under NIM + Light + H_2_O_2_ system. (**b**) DMPO in water. (**c**) DMPO in DMSO.

**Figure 7 molecules-29-05982-f007:**
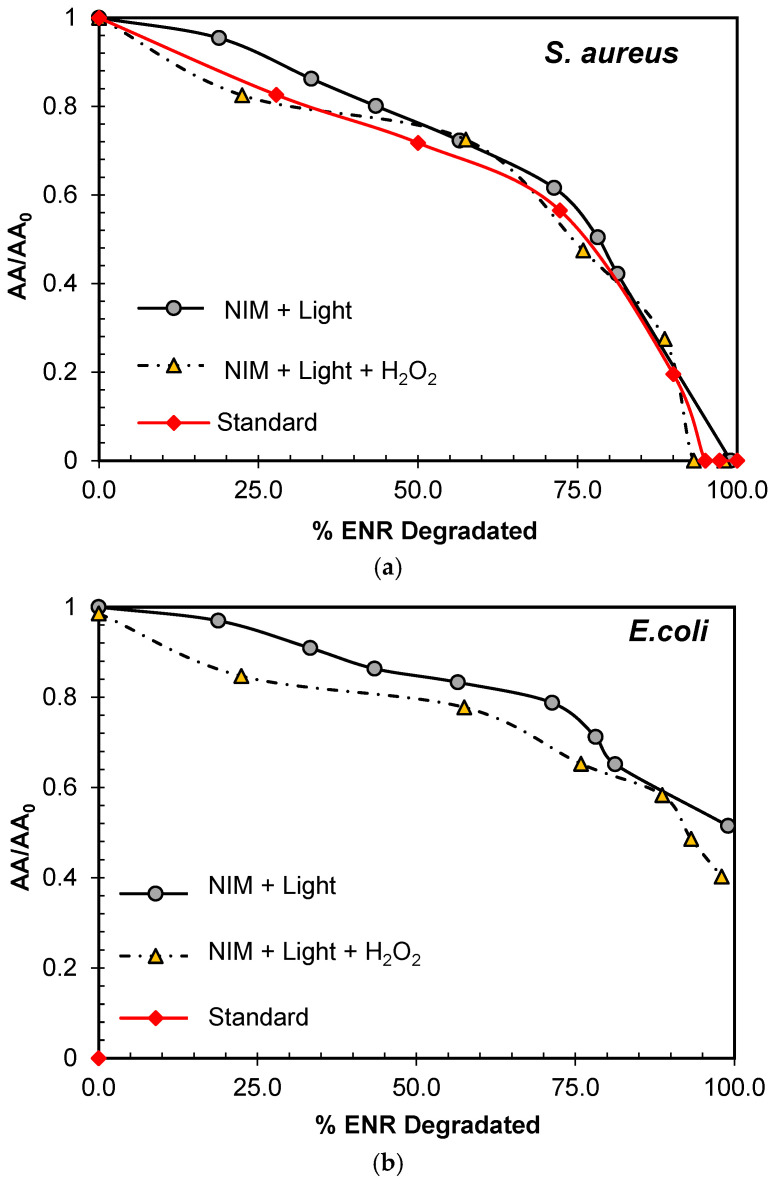
Comparative antimicrobial activity (AA) removal as a function of ENR concentration or elimination by several treatments in distilled water. (**a**) *S. aureus*, (**b**) *E. coli.* Conditions: [ENR] = 0.05 mmol L^−1^; [NIM] = 10.0 mg L^−1^; [H_2_O_2_] = 1.0 mmol L^−1^; pH 6.5.

**Figure 8 molecules-29-05982-f008:**
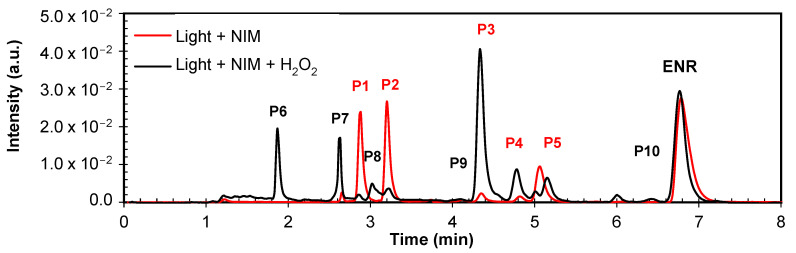
Chromatograms at ~75% degradation of ENR in distilled water under photocatalytic systems (NIM–Light and NIM–Light–H_2_O_2_). Conditions: light 500 W m^−2^; [ENR] = 0.05 mmol L^−1^; [NIM] = 10.0 mg L^−1^; [H_2_O_2_] = 1.0 mmol L^−1^; pH 6.5.

**Figure 9 molecules-29-05982-f009:**
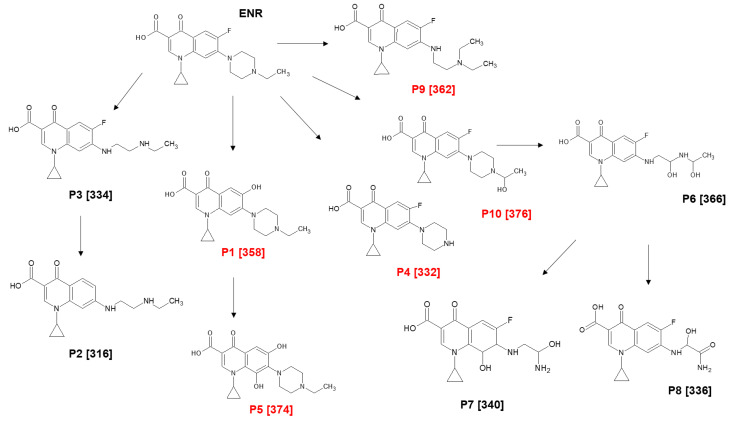
Proposed structures for the primary intermediaries generated by the tested processes.

**Figure 10 molecules-29-05982-f010:**
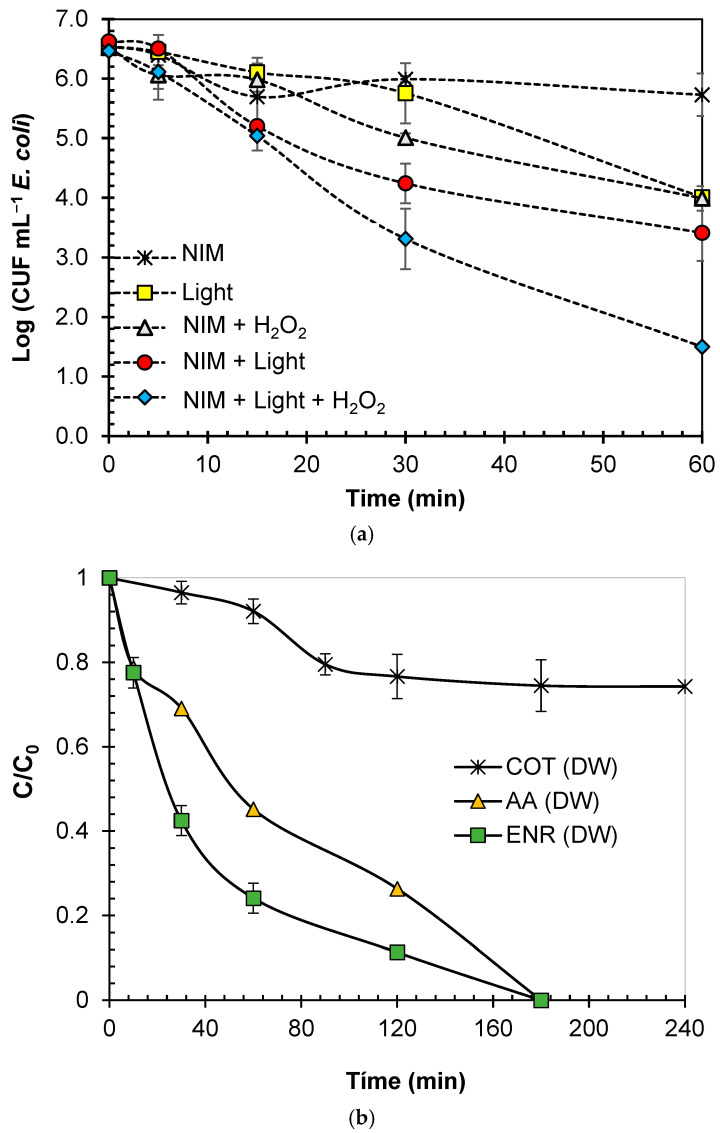
NIM–Light–H_2_O_2_ treatment extent in distilled water. (**a**) Disinfection. (**b**) Mineralization. Conditions: [ENR] = 0.05 mmol L^−1^; [NIM] = 10.0 mg L^−1^; [H_2_O_2_] = 1.0 mmol L^−1^.

**Figure 11 molecules-29-05982-f011:**
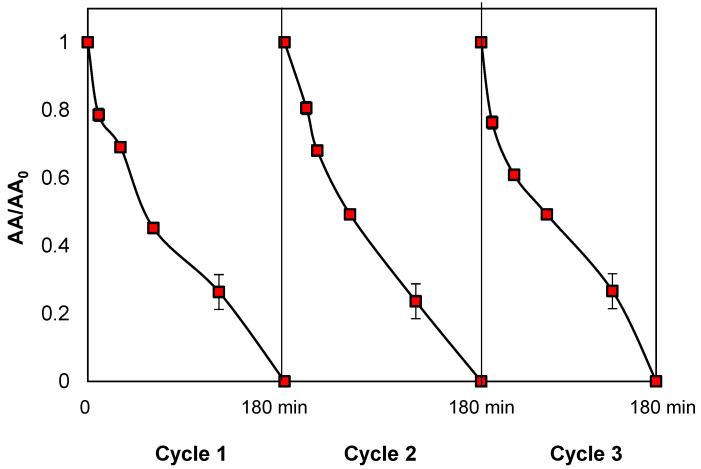
Reusability of catalyst in NIM–Light–H_2_O_2_. Conditions: [ENR] = 0.05 mmol L^−1^; [NIM] = 10.0 mg L^−1^; [H_2_O_2_] = 1.0 mmol L^−1^.

**Figure 12 molecules-29-05982-f012:**
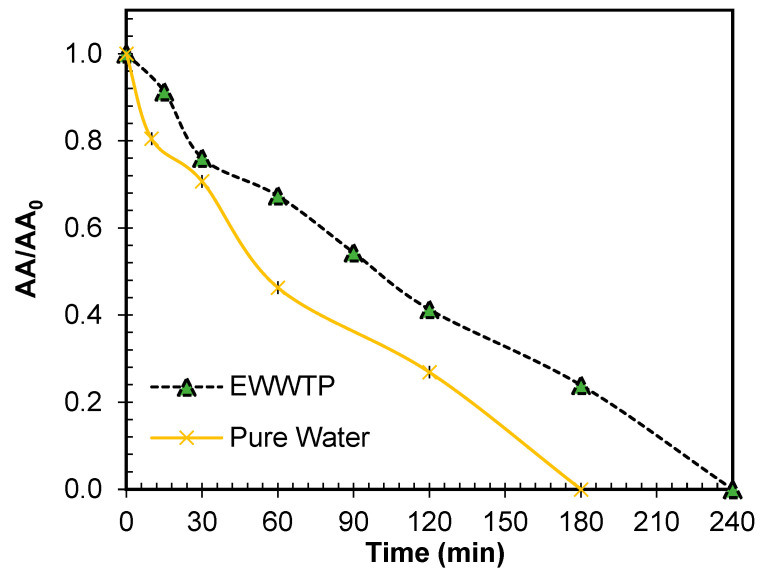
Comparison of ability of NIM–Light–H_2_O_2_ process to remove AA from distilled water (pure water) and effluent of municipal wastewater treatment plant (EWWTP). Conditions: [ENR] = 0.05 mmol L^−1^; [NIM] = 10.0 mg L^−1^; [H_2_O_2_] = 1.0 mmol L^−1^.

## Data Availability

Data will be made available on request to the authors.

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
