# Peer review of "The Photocatalytic Degradation of Enrofloxacin Using an Ecofriendly Natural Iron Mineral: The Relationship Between the Degradation Routes, Generated Byproducts, and Antimicrobial Activity of Treated Solutions"

_molecules, 2024, doi:10.3390/molecules29245982_

Round 1

Reviewer 1 Report

Comments and Suggestions for Authors

In this paper, the photocatalytic ability of natural iron material (NIM) to remove ENR from water was characterized and investigated. The work addresses the current problem of environmental pollution with persistent pharmaceutical compounds. Additionally, as the authors noted, alternatives to conventional wastewater treatment methods are sought, which are not effective. The work is well planned, of very high scientific quality. The authors presented a great number of results, which are well commented on with the references. The work is of very good quality. However, before publication, the authors need to make corrections to supplement several issues. Detailed comments are given below:

·        Introduction - please add information on the sources of pharmaceutical compounds, including antibiotics, into sewage and the environment

·        introduction - please add numerical data if possible, which will draw attention to the importance of the research conducted by the authors. For example, you can write how the consumption of antibiotics is on a global scale, at what concentration levels are the antibiotics identified in environmental matrices

·        2.2. Photocatalytic Capacity of NIM for the Enrofloxacin Degradation and Insights into the Degradation Pathways - Can the authors compare the ENR removal efficiency of the NIM-based photocatalyst system with that of traditional photocatalysts reported in the literature?

·        Fig. 4. c-f - please improve the quality of the figures

·        2.4.1. AA removal considering two representative microorganisms- Two representative microorganisms, S. aureus (Gram-positive bacteria) and E. coli (Gram-negative bacteria), were used in the study. I ask the authors to add a reference to the literature confirming that other authors have also conducted studies on the same indicator organisms.

·        3.1. Reagents - please provide the purity of the chemical reagents used, or their concentration used in the article

·        3.3. Reaction Systems- On what basis did the authors use 0.05 mmol L-1 ENR concentration? Moreover, "The concentration of the antibiotics in the real water was ensured by adding 25 mL of a 0.5 mmol L-1 solution of the contaminant to 240 mL of the real sample" - did the concentration prepared in this way correspond to the ENR concentrations identified in the environment?

·        Chromatographic Analyses - on what basis did the authors identify the transformation products? Were the transformation products identified based on the analysis of characteristic m/z or based on comparing the ms spectra with the NIST database?

·        Conclusions- What are the future research perspectives? Should additional studies be conducted or other reaction systems be tested as alternatives to traditional purification methods? Maybe it would be worth testing other compounds under similar conditions as ENR?

Author Response

Please find attached responses to Reviewer 1 

Reviewer 2 Report

Comments and Suggestions for Authors

The authors have conducted research entitled Photocatalytic Degradation of Enrofloxacin Using an Eco-friendly Natural Iron Mineral: Relationship Among the Degradation Routes, Generated By-Products and Antimicrobial Activity of Treated Solutions. The presented work is novel and has broad interest in the scientific community. Therefore, I recommend publishing it in the Molecules after major revision.

Before being considered for publication, the following adjustments should be made.

1. The novelty of the research article should be highlighted.

2. The research gap should be addressed at the beginning of the abstract.

3. The following work should be considered and cited in the introduction section (page 2, line 54) to give readers a general view: https://doi.org/10.3390/molecules28217326. 

4. The manuscript's hyphen symbol (-1) should replace the minus symbol (–1).

5. Authors should carefully read the manuscript and correct all typos.

6. The resolution of Fig. 1a should be increased. Obtained EDS results should be correlated with another characterisation.

7. Authors commented on XPS results for iron. Authors should also present XPS for oxygen.

8. XRD is a vital tool for materials characterization. Authors should include XRD graphs in the manuscript, not in the supplement material.

9. Figure 1d should be moved from the current position in the manuscript to section 2.2.

10. The resolution of Figure 1e should be increased. One SEM micrograph is enough in the manuscript. Other magnifications should be moved in the supplement material.

11. Resolution of Fig 4c-f, 5b-d, and 6b-c should be improved.

12. Catalytic results should be better explained and correlated with the results of the catalyst's structural, morphological, and optical properties.

13. Authors should put all symbols in italics.

14. Obtained results in the manuscript should be compared with the published literature.

Author Response

Please find attached responses to Reviewer 2.

Reviewer 3 Report

Comments and Suggestions for Authors

The manuscript is an impressive work on the capacities of a natural mineral to be used as a photocatalyst for medical molecules such as antibiotics.

However, since the basic compound is a mixture of various minerals, the structural study of each constituent was essential for a good definition of the composite system.

The announced supplements are not accessible. Figure S1 of the X-ray diffraction analysis is not accessible. However, the crystallographic characterization of the products used is a fundamental step in a scientific study. It must therefore be an integral part of the body of the text.

A lot of technical details are given but some parts would need to be simplified.

Author Response

Please find attached responses to Reviewer 3.

Round 2

Reviewer 2 Report

Comments and Suggestions for Authors

Acceptable for publication in its present form.